# Sociocultural Attitudes towards Appearance, Self-Esteem and Symptoms of Body-Dysmorphic Disorders among Young Adults

**DOI:** 10.3390/ijerph16214236

**Published:** 2019-10-31

**Authors:** Mohammad Ahmadpanah, Mona Arji, Jaleh Arji, Mohammad Haghighi, Leila Jahangard, Dena Sadeghi Bahmani, Serge Brand

**Affiliations:** 1Research Center for Behavioral Disorders and Substance Abuse, Hamadan University of Medical Sciences, Hamadan 6516848741, Iran; m1ahmad2000@gmail.com (M.A.); mona.arji7@gmail.com (M.A.); dr_haghighi_ps@yahoo.com (M.H.); jahangard@umsha.ac.ir (L.J.); 2Department of Educational Sciences, Shiraz Branch, Islamic Azad University, Shiraz 7198774731, Iran; arjeah@gmail.com; 3Center for Affective, Stress and Sleep Disorders, University of Basel, Psychiatric Clinics, 4002 Basel, Switzerland; dena.sadeghibahmani@upk.ch; 4Substance Abuse Prevention Research Center, Kermanshah University of Medical Sciences, Kermanshah 6719851115, Iran; 5Sleep Disorders Research Center, Kermanshah University of Medical Sciences, Kermanshah 6719851115, Iran; 6Isfahan Neurosciences Research Center, Alzahra Research Institute, Isfahan University of Medical Sciences, Isfahan 8174675731, Iran; 7Division of Sport Science, Department of Sport, Exercise and Health, University of Basel, 4052 Basel, Switzerland

**Keywords:** body shape, body dysmorphic disorder, self-esteem, sociocultural attitude towards appearances

## Abstract

*Background:* Beauty and an attractive body shape are particularly important during early adulthood, as both are related to greater mating success, positive social feedback, and higher self-esteem. The media may further influence common features of beauty. We tested whether higher body-dysmorphic disorder (BDD) scores were associated with sociocultural attitudes towards appearance. Additionally, we expected that a link between higher BDD scores and higher perceived media pressure would be mediated by lower self-esteem (SE). *Method:* 350 young Iranian adults (mean age: 24.17 years; 76.9% females) took part in the study. Participants completed questionnaires covering sociodemographic data, sociocultural attitudes towards appearances, and SE, while experts rated participants for symptoms of body dysmorphic disorders. *Results:* Higher BDD scores were associated with higher scores for sociocultural attitudes towards appearance, while SE was not associated with BDD or sociocultural attitudes towards appearance. Higher scores for sociocultural attitudes towards appearance and media pressure predicted higher BDD scores, while SE had no influence. *Conclusion:* Among young Iranian adults, sociocultural attitudes towards appearances and BDD scores, as rated by experts’, were related, while SE was not. The shared variance between symptoms of BDD and sociocultural attitudes towards appearance was low, suggesting that other factors such as mating and career concerns together with social feedback might be more important in explaining symptoms of body dysmorphic disorders.

## 1. Introduction

From an evolutionary point of view, beauty and an attractive body shape are two markers of physical and psychological health, along with the concept of “good genes” and a healthy immune system [1]. Thus, beauty and an attractive body shape reflect the concept of a healthy physiology that is free of genetic disorders and infectious diseases. In this context, Wald [2] observed that thinkers and artists throughout history have believed facial beauty and particularly facial symmetry to be associated with psychological characteristics such as honesty, reliability and trustworthiness. Gordon et al. [3] found, among adolescents, that physical attractiveness was associated with better social integration, less social stigma and higher academic attainment. Likewise, Kirsch et al. [4] concluded in their review of the associations of neuronal processes with human faces and bodies (both static and dynamic), that specific neural areas are responsible for the processing of beautiful faces and bodies. Thus, Kirsch et al. [4] offer a neuronal explanation as to why humans want and like symmetric and beautiful faces and bodies. It follows, therefore, that concepts of beauty should also be found in countries such as Iran, even though for cultural and religious reasons, large parts of a female’s body are covered, including scalp hair, which again from an evolutionary point of view is considered a signal of attractiveness and physical health. 

Next, in earlier studies, Singh [5] found that a lower waist-to-hip ratio in women was associated with a higher mate value, higher perceived fertility and greater attractiveness (“lower” waist-to-hip-ratio means that the waist has a smaller circumference than that of the hip). In contrast, higher waist-to-hip ratios were associated with an increased risk of diabetes, stroke, low fertility and abortion. Importantly, Singh [5] showed that the association between waist-to-hip ratio and female attractiveness was not culture-specific and not inculcated by modern Western fashion dictates or media. Platek and Singh [6] asked male observers to rate the attractiveness of female bodies before and after a surgical intervention to decrease the waist-to-hip ratio. Their results showed that, compared to body shapes before surgical intervention, body shapes after the intervention were rated as more attractive. Significantly, this increase in attractiveness ratings was associated with higher neuronal activation in the bilateral orbitofrontal cortex and sub-cortical structures such as the nucleus accumbens, caudate and putamen. That is, neuronal centers associated with reward and dopaminergic circuits were involved in processing perceptions of attractive body shapes. These findings, along with those summarized in the review by Kirsch et al. [4] and with previous studies of the waist-to-hip ratio [5,6] suggest that the elaboration of beautiful faces and bodies is rooted in specific brain areas related to reward. It follows that the attractiveness of beauty, beautiful faces and bodies are *not* inventions of modern times, and that therefore such mechanisms are *not* culture-specific and not inculcated by modern Western fashion dictates or media. However, as described in more detail below [7,8,9], and unlike previous cultural epochs, today’s media such as fashion magazines, TV and social media have the power to multiply, intensify and exaggerate the universal features of beauty. Specifically, the omnipresence of media disseminating examples of beauty (faces, body shape) appears to impact negatively on individuals’ psychological health. 

If we consider that the appraisal of an ideal body shape is (1) neuronally and (2) evolutionarily rooted, and (3) understood as a marker of both physical and psychological health to (4) enhance mating success and (5) social integration, it is not surprising that young adults in particular should feel under pressure to match expectations of beauty and ideal body shape. In addition, these ideals of beauty and body shape can be further reinforced by social media. As an example, three years after the introduction of television into the Fiji Islands in 1998, Becker [7] and Becker et al. [8] investigated the impact of TV with Western programming on Fijian female adolescents (mean age: 16.9 years). Becker [7] and Becker et al. [8] reported the following observations: Firstly, adolescents indicated that the attitudes of characters presented in TV dramas had become models of adolescents’ behavior. Secondly, participants also reported (a) increases in preoccupation with weight and body shape, (b) increases in practices such as purging behavior to control weight, and (c) greater indications of disordered eating behavior. In addition to TV, social media include social networking sites (SNSs) such as Facebook^®^ and Instagram^®^ [9]. In their own systematic review of 20 studies, Holland and Tiggemann [9] found that the use of SNSs focusing on body shape and beauty was associated with greater body-image dissatisfaction and more disordered eating behavior. Furthermore, specific SNS activities (viewing and uploading photos; seeking feedback via status updates; appearance-based social comparison) appeared to be particularly associated with body image dissatisfaction and eating-disordered behavior. Similarly, Meretuk et al. [10] reported that apparently up to three million people follow Pamela Reif’s channel “Strong & Beautiful”^®^ on Instagram^®^, and that beauty ideals and presentations of this kind, spread in the media, often reinforce self-doubt and insecurity among adolescents, as assessed in a sample of 730 adolescents aged 13 to 20 years. 

To conclude, media such as SNSs might promote particular standards of beauty and body shape, which in turn may further increase the gap between individual and socially idealized body shape. Accordingly, the first aim of the present study was to investigate the associations between symptoms of body dysmorphic disorders as a proxy for dissatisfaction with body shape and appearance and sociocultural attitudes towards appearance in a sample of young adults in Iran. 

Additionally, one might anticipate that symptoms of body dysmorphic disorders as proxies for issues around body shape [11] will be associated with lower self-esteem [12,13,14]. This expectation follows from the observation that self-esteem, particularly among females, is related to others’ feedback on body shape and behavior [15,16]. It follows that negative feedback and a gap between socially expected body shape and own current body shape might be associated with lower self-esteem. However, results from previous studies suggest that the association between self-esteem and body dysmorphic disorders (understood as proxies for dissatisfaction with one’s own body and body shape) is not linear or straightforward. As also noted by Cerea et al. [13], Hartmann et al. [12], and Rosen and Ramirez [17], lower levels of self-esteem have been observed among samples of individuals with body image disorders (eating disorders and body dysmorphic disorders) than among healthy controls. Buhlmann et al. [18] assessed implicit attitudes of self-esteem among individuals with body dysmorphic disorders, and observed that such individuals had lower implicit self-esteem than either individuals with subclinical body dysmorphic disorders or healthy controls. In contrast, Bohne et al. [19] were unable to find differences in self-esteem between healthy controls and individuals with body dysmorphic disorders. A similarly inconsistent pattern has emerged from investigations of the links between self-esteem and eating-disordered behavior [20], with some studies finding an association between low self-esteem and eating-disordered behavior [21,22,23] and others not [20,24]. Some studies have even reported higher scores for self-esteem to be associated with eating-disordered behavior [25,26,27]. In the context of eating disorders, it is important to note the high overlap between symptoms for body dysmorphic disorders and those for eating disorders [28]. For example, Dingemans et al. [29] showed that of 158 patients seeking treatment for eating disorders more than half also reported body dysmorphic disorders. Phillipou et al. [30] noted that individuals with anorexia (as a proxy for eating disorders) reported distortions in the experience and/or satisfaction with one’s own body, and the anxiety associated with their body image. Accordingly, Phillipou et al. [30] proposed to re-classify anorexia nervosa under a new category of body image disorders. In doing so, body image would become the dominant feature, such as in bulimia nervosa and, most importantly, in body dysmorphic disorder (BDD).

To summarize, while it appears plausible that lower self-esteem and higher scores for body dysmorphic disorders will be associated, results are mixed. Accordingly, the second aim of the present study was to investigate the association between self-esteem on the one hand and symptoms of body dysmorphic disorders and sociocultural attitudes towards appearance on the other in a sample of young adult Iranians. The aspiration of the present study is therefore to shed further light on the association between symptoms of body dysmorphic disorders and sociocultural attitudes towards appearance among young Iranian adults. Note that in the case of females and for cultural and religious reasons, Iranian adults are less able to display body shape and parts of the body; more specifically, from the view point of evolutionary psychology [1,31], showing (slim) body shape and strong, shiny and long hair might be considered cues of females’ younger age and physical health, which might be deemed attractive and therefore sexually arousing. Another significant feature of the present study is its inclusion of the dimension of self-esteem. As noted above, the importance of self-esteem for body satisfaction is uncertain; accordingly, the present study aims to determine whether, and if so to what extent, self-esteem is associated with dissatisfaction with one’s own body and body shape.

The following two hypotheses and three research questions were formulated. Firstly, following others [7,8,9,10], who showed that higher body image dissatisfaction was related to media use related to body shape and beauty, we anticipated that higher scores for body dysmorphic disorders, as rated by experts, would coexist with higher scores for sociocultural attitudes towards appearance. Secondly, and again following others [9,10,32] we anticipated that individuals with a high risk of body dysmorphic disorders would also have higher scores for sociocultural attitudes towards appearance. Our first exploratory research question concerned the extent if, and if so, to which expert rated symptoms of body dysmorphic disorders and sociocultural attitudes towards appearance would be related to self-esteem. This research question was exploratory, because previous studies have yielded contradictory results [12,13,14,17,18,19]. Our second exploratory research question concerned whether self-esteem mediated associations between symptoms of body dysmorphic disorders and sociocultural attitudes towards appearance. The third exploratory research question sought to identify the dimensions of sociocultural attitudes towards appearance that might best predict symptoms of body dysmorphic disorders. 

To address these hypotheses and research questions, a sample of 350 young Iranian adults completed questionnaires on self-esteem, and sociocultural attitudes towards appearance, while experts rated participants’ severity of body dysmorphic disorders (see details below).

## 2. Methods

### 2.1. Procedure

Students of the Hamadan University of Medical Sciences (Hamadan, Iran) were approached to participate in the present cross-sectional study. Eligible participants were informed about the aims of the study and the confidential and anonymous data handling. Thereafter, they signed a written informed consent and completed a series of questionnaires covering sociodemographic data, sociocultural attitudes towards appearance (see below), and self-esteem (see below), while experts rated their symptoms of body dysmorphic disorders (see below for a thorough description). The ethics committee of the Hamadan University of Medical Sciences (HUMS; Hamadan, Iran) approved the study (IR.UMSHA.REC.1395.261), which was conducted in accordance with the rules laid down in the seventh and current edition (2013) of the Declaration of Helsinki.

### 2.2. Sample

During the spring term 2018 (May to June) all second-semester students of the faculties of medicine and psychology were approached to participate in the present cross-sectional study. Faculty members advertised the study during their classes. Those students who agreed to participate in the study completed the booklet over a period of 15–20 min after the last session of the day; afterwards, experts’ rated participants’ symptoms of body dysmorphic disorders. No specific inclusion and exclusion criteria were formulated, except for the following: (a) age at least 18 years; (b) signed written informed consent; (c) currently student of medicine or psychology in the second semester; (d) willing and able to complete the questionnaires. Participants were not reimbursed for their participation.

### 2.3. Tools

#### 2.3.1. Sociodemographic Data

Participants reported their age, gender, marital status (single; married), and current educational level (bachelor, master, PhD-program).

#### 2.3.2. Body Dysmorphic Disorder (BDD)

To assess symptoms of body dysmorphic disorder (BDD), experts (trained psychiatrists and clinical psychologists) employed the Yale-Brown Obsessive-Compulsive Scale for Body Scale, modified for Body Dysmorphic Disorder (BDD-YBOCS) [33]. Rabiei et al. [34] translated the questionnaire into Farsi/Persian; psychometric properties of the Farsi questionnaire were satisfactory. The questionnaire consists of 12-items designed to rate severity of body dysmorphic disorder (BDD) in people showing excessive preoccupation and subjective distress with physical appearance. The first 10 items assess excessive preoccupation, obsessions, impairment of global functioning, subjective distress, and compulsive behaviors associated with dissatisfaction with physical appearance. Typical items are: “I spent time thinking about body defects”; “I spent time in activities related to body defects”; “I feel distressed with thoughts about body defects”. Items 11 and 12 assess insight and avoidance, respectively. Answers are given on five-point rating scales ranging from 0 (=no symptoms) to 4 (=extreme BDD symptoms), with higher scores therefore reflecting more marked BDD symptoms (maximum score: 48). In addition to the overall score, we followed Rabiei et al [34] in clustering items into the following four dimensions: Preoccupations (#1–5); Repetitive behavior (#6–10); Avoidance (#11); Insight (#12). Following Phillips et al. [35], scores of 20 and more points were treated as indicative of body dysmorphic disorders (Cronbach’s alpha in the present study = 0.89). 

#### 2.3.3. Sociocultural Attitudes towards Appearance

To assess sociocultural attitudes towards appearance, participants completed the Sociocultural Attitudes toward Appearance Questionnaire (SATAQ-3) [36]. Mousazadeh et al. [37] translated the questionnaire into Farsi/Persian, achieving satisfactory psychometric properties. The questionnaire consists of 30 items. Typical items are: “I would like my body to look like the models who appear in magazines.”; “I compare my body to the bodies of people who appear in magazines.”; “I compare my body to that of people in “good shape””. Responses are given on five-point scales with the anchor points 1 (=completely disagree) to 5 (=totally agree), with higher scores reflecting a stronger tendency to adapt appearance toward sociocultural expectations. Following Warren et al. [38], four subscales were derived as follows. The Internalization-General subscale (9 items) assesses the acceptance of a thin body ideal; the Internalization-Athlete subscale (5 items) measures the internalization of media influences related to the achievement of an athletic physique. The Information subscale (9 items) refers to the respondent’s belief that the media are good sources of information about appearance and fashion. The Pressures subscale (7 items) assesses a subjective sense of feeling pressure from the media to modify one’s physical appearance (Cronbach’s alpha in the present study = 0.84).

#### 2.3.4. Self-Esteem 

To assess self-esteem, the Rosenberg Self-Esteem Questionnaire was employed [39]. Shapurian et al. [40] translated the questionnaire into a Farsi/Persian version with satisfactory psychometric properties (Cronbach’s alpha > 0.84; one-factor solution). The questionnaire consists of 10 items, and typical items are: “On the whole, I am satisfied with myself.”; “I feel that I have a number of good qualities.”; or “I feel I do not have much to be proud of.”. Answers are given on 4-point rating scales ranging from “strongly agree” to “strongly disagree”, with some items reverse coded, and with higher scores reflecting higher self-esteem (Cronbach’s alpha in the present study = 0.87). 

### 2.4. Statistical Analysis

Preliminary calculations: We explored whether marital status or educational level systematically differed between male and female participants via a series of X^2^-tests. All X^2^-values were < 1.00, *p* > 0.40. On the basis of one-way ANOVAs, we examined whether gender, marital status, or educational level systematically biased dimensions of body dysmorphic disorders, self-esteem, or sociocultural attitudes towards appearance. All F’s were < 1.00, *p* > 0.50. Accordingly, gender, marital status, and current educational level were not introduced as possible confounders. 

Pearson’s correlations were computed to determine associations between symptoms of body dysmorphic disorders, sociocultural attitudes towards appearance, and self-esteem. Next, partial correlations were calculated for symptoms of body dysmorphic and sociocultural attitudes towards appearance, controlling for self-esteem. Participants were then split into two groups, one at low risk of BDD (BDD scores ≤ 19) one at high risk of BDD (BDD scores ≥ 20). A series of t-tests was performed with low vs. high risk of BDD as independent variable and sociocultural attitudes toward appearance and self-esteem as dependent variables. Last, we determined via a multiple regression which dimensions of sociocultural attitudes towards appearance predicted body dysmorphic disorder scores. Durbin-Watson coefficients are reported to indicate if residuals were independent. In addition, R and R^2^ are reported to indicate whether multiple regression models sufficiently explained the dependent variables. Predictors were excluded from the equation if they do not reach statistical significance.

The nominal level of significance was set at alpha ≤ 0.05. All statistical computations were performed with SPSS^®^ 25.0 (IBM Corporation, Armonk, NY, USA) for Apple^®^ Mac^®^.

## 3. Results

### 3.1. Study Sample

Of 378 individuals approached, 350 (mean age: 24.17 years; range: 19–27 years) agreed to take part in the study; of these, 269 (76.9%) were females and 81 (23.1%) were males. In addition, 239 (68.3%) were single and 111 (31.7%) were married. Seventy-five (21.4%) were studying nursing or medicine, and 275 (78.6%) were studying psychology; 270 (77.1%) were at bachelor or master level, five (1.4%) were doctoral students, and 75 (21.4%) did not report their current level.

### 3.2. Associations between Body Dysmorphic Disorders, Sociocultural Attitudes towards Appearance, and Self-Esteem

Table 1 reports the descriptive statistics and correlation coefficients for body dysmorphic disorders, sociocultural attitudes towards appearance, and self-esteem. 

Scores for body dysmorphic disorders, as rated by experts, consisted of the following dimensions: Preoccupations; repetitive behavior; avoidance, insight, and a total score. Scores of sociocultural attitudes towards appearance consisted of the following dimensions: Internalization general; Internalization of athletic and sport figure; Pressures, and Information. As shown in Table 1, the expert ratings of body dysmorphic disorders were associated with higher scores for sociocultural attitudes towards appearance. Specifically, preoccupations with physical appearance and repetitive behavior to cope with physical appearance were associated with internalization of sociocultural attitude towards appearance, pressures to conform to thin body shapes and internalization of an athletic figure. Avoidance was not associated with dimensions of sociocultural attitudes towards appearance. Higher scores of insights were associated with higher scores of internalization of athletic and sport figure and the SATAQ total score. Correlations between self-esteem and dimensions of body dysmorphic obsessions, and sociocultural attitudes towards appearance were trivial to small and non-significant. The component dimensions of body dysmorphic disorders were moderately to highly intercorrelated, as were the component dimensions of sociocultural attitudes towards appearance.

### 3.3. Low and High Risk of Body Dysmorphic Disorders and Sociocultural Dimensions of Appearance 

Table 2 provides the descriptive and inferential statistical indices for participants at respectively low (BDD scores < 20; n = 316) and high risk of body dysmorphic disorders (BDD scores ≥ 20; n = 34). 

As regards sociocultural attitudes towards appearance, participants at high risk of body dysmorphic disorders had higher scores for Internalization general, Internalization of athletic and sport figure, and pressures (medium effect sizes). For Information and the total score, effect sizes were trivial or small. 

### 3.4. Self-Esteem, Body Dysmorphic Disorders and Sociocultural Attitudes towards Appearance

Table 1 also gives the correlations between self-esteem, expert rated body dysmorphic disorders and sociocultural attitudes towards appearance. Correlations coefficients were low and non-significant. Likewise, participants with a low risk of BDD did not have a higher self-esteem score, than those with a high risk of BDD (see Table 2).

### 3.5. Correlations between Body Dysmorphic Disorders and Sociocultural Attitudes towards Appearance, Controlling for Self-Esteem

Correlations coefficients between body dysmorphic disorders and sociocultural attitudes towards appearance were unchanged when self-esteem was introduced as co-variate. 

### 3.6. Dimensions of Sociocultural Attitudes towards Appearance and Self-Esteem as Predictors of Body Dysmorphic Disorders 

Table 3 reports the results of the multiple regression analysis with total expert rated body dysmorphic disorders score as dependent variable and dimensions of sociocultural attitudes towards appearance and self-esteem as predictors. 

Together the predictors explained 12% of variance in the body dysmorphic disorders total score (R = 0.348; R^2^ = 0.120; Durbin-Watson coefficient: 1.860). Higher scores for sociocultural attitudes towards appearance of Internalization general and Pressures predicted higher BDD scores, while self-esteem, Information and Internalization of athletic figure were excluded as predictors, as they did not reach statistical significance. 

## 4. Discussion

The key findings of the present study were that among a sample of 350 young Iranian adults, body dysmorphic disorders, as rated by experts, were associated with sociocultural attitudes towards appearance, while self-esteem appeared to be unrelated to either body dysmorphic disorders or sociocultural attitudes towards appearance. The present findings are novel and add to the current literature in an important way in that the pattern of results confirms the importance of bodily appearance in relation to sociocultural attitudes towards appearance in a sample of young adult Iranians, while self-esteem appeared to be statistically unrelated. Furthermore, the findings are important, as they showed that dimensions of body dysmorphic disorders as a proxy of low satisfaction with one’s body shape was only poorly explained (12% of the variance) by dimensions of sociocultural attitudes towards appearance. It follows that a larger part of dimensions of body dysmorphic disorders should be explained by further, but latent and unassessed, psychological dimensions.

Two hypotheses and three research questions were formulated and each of these is considered in turn.

Our first hypothesis was that the risk of body dysmorphic disorders would be associated with sociocultural attitudes towards appearance, and this was confirmed. The present results are thus consistent with the findings of previous studies [9,10]. For two reasons, however, the present results expand upon previous findings. Firstly, unlike previous studies, symptoms of body dysmorphic disorders were rated by experts, and this approach reduced the risk of biased results. Secondly, the association was found to hold for young adults in Iran, where for cultural reasons showing face, hair and body shape are restricted to a minimum in public places and on TV channels, at least for females. Nevertheless, the results also show that concerns about beauty and body shape appear to be related to the influence of media such as Social Networking Sites, but only partially; as Table 1 shows, Avoidance was not associated at all, and Insight was only modestly associated with internalization of athletic and sport figure, and the total score of the questionnaire. Further, sociocultural attitudes towards appearance explained 3.72% to 10.89% of the variance in body dysmorphic disorder scores. Thus, 89.11% to 96.28% of the variance in these scores is not accounted for by these attitudes. While the data available from the study is unable to shed light on other factors accounting for this variance, the following speculations are advanced. Firstly, the feeling that one has an imperfect body might be shaped by feedback from peers and family members [10,15,16,41,42]; indeed, the quality of social feedback, or biased perception of such feedback, appears to have an impact on our psychological health. Thus, Ding et al. [43] noted that perceived chronic social adversities may lead to symptoms of stress, depression, and anxiety, and it seems possible that such perceptions could also produce the symptoms of body dysmorphic disorders. Furthermore, at least among adolescents, physical attractiveness is associated with better social integration, less social stigma and higher academic attainment [3]. Secondly, young adults, compared to adolescents and older adults, are under greater pressure to find a partner and to achieve success in their professional lives. “Success” in mating and career appear to be related to attractiveness [44], and it is therefore not surprising that media are not the exclusive sources of satisfaction with body appearance. 

Our second hypothesis was that individuals scored by experts on the BDD-YBOCS at or above a cut-off of 20 points would also have higher scores for sociocultural attitudes towards appearance. This expectation was not fully supported, as the total score of the SATAQ and the dimension Information did not statistically significantly differ between the two groups. The pattern of results here is more consistent with our conclusions regarding the first hypothesis; symptoms of body dysmorphic disorders and sociocultural attitudes towards appearance are moderately related, as satisfaction or dissatisfaction with appearance appears to be driven by other factors such as social feedback.

Our first and second exploratory research questions concerned the extent to which symptoms of body dysmorphic disorders and sociocultural attitudes towards appearance were related to self-esteem. We found that self-esteem was unrelated to either body dysmorphic disorders or sociocultural attitudes towards appearance (see Table 1). It was therefore going to be the case that self-esteem would not emerge as a confounder of any association between symptoms of body dysmorphic disorders and sociocultural attitudes towards appearance (answering the second research question). The present pattern of results fits well with the mixed and confused pattern of results reported so far as regards the link between self-esteem and body satisfaction. Some researchers have found that low self-esteem is related to body dysmorphic disorders and to sociocultural attitudes towards appearance [12,13,14,17,18]. However, Bohne et al. [19] were unable to confirm such a pattern, and our own findings are more consistent with those of these latter authors. While we have no direct evidence regarding underlying cognitive-emotional processes, we believe that the present pattern of results has parallels with relations observed between self-esteem and eating-disordered behavior: While it seems plausible that eating disorders would be related to low self-esteem, the evidence for such a relation is quite inconsistent; some studies have reported a positive association, some a negative association, and some have found no association between eating disorders and self-esteem [20,21,22,23,24].

The third exploratory research question concerned the dimensions of sociocultural attitudes towards appearance that might best predict symptoms of BDD. As shown in Table 3, higher scores for sociocultural attitudes towards appearance of general information and media pressure predicted higher scores for BDD, while attitude towards appearance of an athletic figure, and media as information did not reach statistical significance. Again, the data available to us cannot shed any direct light on the cognitive-emotional mechanisms underlying these relations. This holds also true as regards the observation that Internalization-General (that is: the acceptance of a thin body ideal) was descriptively more predictive than Pressures from media (that is: the subjective sense of feeling pressure from the media to modify one’s physical appearance). However, we also caution that the statistically significant results should not be overstated given that the predictors explained only 12.6% of the variance in the body dysmorphic disorder scores. To put it the other way around: unassessed factors account for 87.4% of the variance in body dysmorphic disorders scores, and again we would point out that factors such as social acceptance, and expectations about mating and professional success may well be more powerful factors. 

Despite the novelty of our findings, several limitations warrant against their overgeneralization. Firstly, except for the assessment of dimensions of body dysmorphic disorders, we relied on self-ratings, while a thorough psychiatric interview [45] would have helped to identify participants with psychiatric issues such as major depressive disorders, personality disorders, eating disorders, and substance use disorders. Secondly, symptoms of depression have been observed among individuals with body dysmorphic disorders [12,13] and with anorexia nervosa [12], and in these studies [12,13], symptoms of major depressive disorders explained the association between higher scores for bodily dissatisfaction and self-esteem. Likewise, perfectionism [12,13] and obsessive-compulsive disorders [13] as dysfunctional personality traits were identified among individuals with body dysmorphic disorders. Latent but unassessed physiological (e.g., metabolic diseases), anthropometric (e.g., BMI) and psychological characteristics (e.g., academic ambition) might have biased the present pattern of results. Thirdly, a key definition of body dysmorphic disorders [11,14] is the gap between individuals’ self-perceptions of their body and others’ view of the individual’s body. Accordingly, it would have been informative to know what other people such as friends, classmates, sports colleagues, neighbors, and family members thought about the participants’ body shape and beauty. The so-called “one-with-many” procedure as extensively described in Holtzman and Strube [46] would have allowed the comparison of individuals’ self-perceptions of body shape with the perceptions of others, and would have reduced the rating bias inherent in any single source of data. Fourthly, a longitudinal study design would have allowed the identification of causal directions. Next, from a cognitive point of view, it would have been interesting to have had more information about underlying cognitive-emotional processes that might explain individual differences in susceptibility to media influences. As Table 1 shows, the correlations between symptoms of body dysmorphic disorders and sociocultural attitudes towards appearance ranged between r = 0.061 and 0.33, indicating that sociocultural attitudes explained from 3.72% to 10.89% of the variance in body dysmorphic disorders, leaving 89.11% to 96.28% unexplained. A possible solution might have been to assess other factors such as peer expectations and family expectations, as proposed in the Tripartite Influence Model on body image [47]. Penultimately, the options for managing one’s body shape and more particularly keeping it lean are basically either to exercise more, or to reduce calorie-intake, or to combine these. Future studies in the field of symptoms of body dysmorphic disorders and sociocultural attitudes towards appearance should therefore assess both physical activity and diet. Lastly, a possible source of bias might be the unbalanced gender ratio, along with sampling from a university student population; it is possible that in samples with lower educational levels or lower socioeconomic status another pattern of results might have been observed. 

## 5. Conclusions

Among a sample of Iranian young adults, dimensions of body dysmorphic disorders and sociocultural attitudes towards appearance were associated, while self-esteem was unrelated. It is likely that further factors such as social feedback and concerns about mating and professional success might have a stronger impact on satisfaction with appearance and body shape. Nonetheless, young adults with symptoms of body dysmorphic disorders might need counseling against the possibility that notions about appearance, body shape and others’ expectations regarding social behavior and success become dysfunctional. Finally, as the present findings indicate when a young adult shows signs of body dysmorphic disorders, this cannot be attributed to low self-esteem. 

## Figures and Tables

**Table 1 ijerph-16-04236-t001:** Overview of correlation coefficients between body dysmorphic disorders (experts’ ratings), sociocultural attitude towards appearance, and self-esteem.

					Dimensions							
		1	2	3	4	5	6	7	8	9	10	11
	Body dysmorphic disorders											
1	Preoccupations	-	0.71 ***	0.24 **	0.37 ***	0.93 ***	0.07	0.31 ***	0.25 ***	0.25 ***	0.31 ***	0.05
2	Repetitive behavior		-	0.29 ***	0.32 ***	0.92 ***	0.04	0.20 ***	0.18 **	0.20 **	0.26 ***	−0.02
3	Avoidance			-	0.23 ***	0.28 ***	−0.09	0.08	0.02	0.00	.00	0.08
4	Insight				-	0.38 ***	0.01	0.11	0.07	0.16 *	0.12 **	−0.03
5	Total score					-	0.06	0.33 **	0.24 **	0.24 **	0.31 *	−0.04
	Sociocultural Attitude towards Appearance											
6	Information						-	0.21 ***	0.16 **	0.14 **	0.63 ***	−0.04
7	Internalization general							-	0.39 ***	0.50 ***	0.80 ***	0.12 *
8	Pressures								-	0.42 ***	0.61 ***	0.06
9	Internalization of athletic and sport figure									-	0.68 ***	0.15 *
10	Total score										-	0.14 *
11	Self-esteem											-
M (SD)		5.92 (3.16)	6.42 (3.13)	1.82 (0.26)	1.01 (0.98)	12.34 (5.81)	28.39 (7.20)	23.15 (7.38)	18.92 (3.67)	13.78 (4.46)	84.24 (15.73)	76.17 (25.69)
Cronbach’s alpha		0.81	0.85	-	-	0.89	0.82	0.81	0.86	0.85	0.84	0.83

Notes: * = *p* < 0.05; ** = *p* < 0.01; *** = *p* < 0.001. Scores of body dysmorphic disorders are experts’ ratings.

**Table 2 ijerph-16-04236-t002:** Descriptive and inferential statistical indices of age, body dysmorphic disorders (experts’ ratings) and sociocultural attitudes towards appearance, and self-esteem, separately for individuals with low body dysmorphic disorders and high body dysmorphic disorders.

	Groups		Statistics	
	body dysmorphic disorders			
	Low	High	t-tests	Cohen’s d
N	316	34		
	M (SD)	M (SD)		
Age (years)	24.26 (3.75)	23.59 (3.29)	t(348) = 1.00	0.31 (S)
Body dysmorphic disorders				
Preoccupations	5.30 (2.55)	11.65 (2.56)	t(348) = 13.78 ***	2.48 (L)
Repetitive behavior	5.81 (2.53)	12.09 (2.39)	t(348) = 13.83 ***	2.55 (L)
Avoidance	0.63 (0.74)	2.59 (2.36)	t(348) = 5.22 ***	1.12 (L)
Insight	0.92 (1.26)	1.88 (1.23)	t(348) = 4.27 ***	0.77 (M)
Total score	11.11 (4.48)	23.74 (4.24)	t(348) = 15.70 ***	2.89 (L)
Sociocultural Attitude towards Appearance				
Internalization general	22.76 (7.33)	26.79 (6.92)	t(348) = 3.06 **	0.56 (M)
Internalization of athletic and sport figure	13.55 (4.50)	15.85 (3.21)	t(348) = 2.88 **	0.52 (M)
Pressures	18.73 (3.50)	20.79 (4.62)	t(348) = 3.10 **	0.51 (M)
Information	28.51 (7.24)	27.24 (6.81)	t(348) = 0.98	0.18 (T)
Total score	83.55 (15.46)	90.68 (16.99)	t(348) = 2.53 *	0.45 (S)
Self-esteem	76.28 (25.54)	75.15 (27.44)	t(348) = 0.25	0.05 (T)

Notes: * = *p* < 0.05; ** = *p* < 0.01; *** = *p* < 0.001; T = trivial effect size; S = small effect size; M = medium effect size; L = large effect size.

**Table 3 ijerph-16-04236-t003:** Multiple linear regression with Body Dysmorphic Disorders (total score) as dependent variable, and Sociocultural Attitudes towards Appearance, and self-esteem as predictors.

Dimension	Variables	Coefficient	Standard Error	Coefficient β	t	p	R	R^2^	Durbin-Watson Coefficient
Body dysmorphic disorders	Intercept	3.60	1.577	-	2.28	0.023	0.348	0.121	1.86
	Internalization general	0.222	0.043	0.283	5.16	0.000			
	Internalization of athletic and sport figure	0.099	0.079	0.076	1.25	0.213			
	Pressures	0.190	0.087	0.120	2.18	0.029			
	Information	0.015	0.0042	−0.019	−0.363	0.717			
	Self-esteem	−0.002	0.012	−0.010	−0.199	0.842

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
