# Peer review of "Sociocultural Attitudes towards Appearance, Self-Esteem and Symptoms of Body-Dysmorphic Disorders among Young Adults"

_ijerph, 2019, doi:10.3390/ijerph16214236_

Round 1

Reviewer 1 Report

Important study, well-done.

Three suggestions:

1)  Review of literature might include work of Anne Becker, who studied changes in body attitudes in Polynesia before and after arrival of television.

2)  Early in paper, should identify Iran as locus of study.

3)  Polishing of English in paper could include revision of "higher" used several times - clarify whether reference is to higher score, greater intensity, or higher prevalence.

Author Response

Please see the detailed point-by-point-response.

Reviewer 2 Report

This study sought to identify what sociocultural attitudes are related to Body Dysmorphic disorders in a sample of graduate and undergraduate Iranian students. The authors also wanted to explore the role of self-esteem in explaining these relationships. The study has a large subject pool, and the data have the potential to explore some interesting cross-cultural questions. Unfortunately, the way the paper is outlined now, the Intro does not explain the significance of the study, nor the rationale for the hypotheses. The use of the YBOCS-BDD is unclear – this is a clinical instrument, not really for the general population. This renders all analyses suspect. The subscales of the SATAQ were not clearly articulated, making it unclear what was being assessed. The data analyses did not clearly follow the hypotheses. All these issues made the Conclusions suspect. I articulate all these issues in detail below.

I would like to offer the authors some suggestions for reworking the paper. You have a unique opportunity to explore how a culture that values modest public presentations of female bodies (head coverings, shapeless garments) can still have body dysmorphia and Westernized views of idealized body shape/size. You want to make the claim that a low ratio of waist to hip size is a universal belief and evolutionarily desirable for health and procreation. I would dispute this to some extent (I’m not a fan of evolutionary arguments in general). Nevertheless, there are cultural variations (even in the research you cite) in the size of that ratio, and to some extent this may reflect the general size/shape of the women living within a culture (e.g., the generally smaller, thinner women in Asian cultures may have higher ideal ratio – waist and hips are both small – compared to women in other cultures). The link I sent for the SATAQ measure is a cross-cultural study of this instrument in the US. You might compare your data to theirs to see how women in Iran view themselves on these measures (yes, eliminate the men in your study; the “thin ideal” is really a female problem). Also, dig into your data to find out: what parts of their bodies are Iranian women unsatisfied with (from BDD)? Do they have a “thin ideal?” What media do they look at (SATAQ) and which influences them the most? It may be the case that Iranian women – at least those that are highly educated - are less dissatisfied with their bodies and view media images of women less often than Westernized women. Indeed, dressing modestly may take the pressure off of these women to realize some idealized body type. In sum, you have a lot of good data here; you need to recraft it to ask more relevant and interesting questions.

General comment – using APA style?

Need comma before “and” in a string of ideas (including the title) Punctuation goes inside quotation marks

Line 22 – no comma after “We tested”

Line 41/42 – Revise to: “. . . and a healthy physiology that is free of genetic disorders or infectious diseases.

Line 43 – “throughout history” – delete “the”

Lines 45/46 – Gordon et al. showed that among adolescents, physical attractiveness was associated with greater social integration, less social stigma . . .

Line 50 – Liking and wanting? What?

Line 51 – add “in women” after the word “ratio”

Line 52 – “. . . higher perceived fertility, and more female attractiveness”

Line 62 – Divide into 2 sentences: “. . . caudate, and putamen. That is, neuronal centers  . . “

Line 72 – “. . .it is not surprising that young adults . . .”

Lines 80-82 – This is a cultural reference I am not familiar with. It does not seem to support the argument that watching SNSs leads to lower body image.

Lines 83-87 Not sure I follow your logic. The beginning of the paper lays the groundwork that beauty has universal definitions – facial symmetry, low WTH ratio – and that the appeal of these features is rooted in neuronal responses to them. The paragraph in Lines 70-82 then makes a claim that SNSs may contribute to lower body image and eating disorders. Your first aim, then, looks at BDD and sociocultural attitudes towards appearance. Not sure I understand why there would be a relationship based on your premises. What sociocultural attitudes do you mean? Facial symmetry and low WTH ratio? This aim is independent of media viewing.

You will need to define what you mean by “sociocultural attitudes” and how the concept fits in with the discussion of universal beauty that has an evolutionary basis.

Lines 88-90 – Not clear what you are saying, especially in the “above all” part.

Line 93 – comma after Ramirez [13]

Line 95 – delete “always” – “as compared to . . .”

Line 97 – same as line 95; also – what is “implicit self-esteem?”

Line 100 – no comma after “appeared”

Lines 109-113 I am not clear how hypotheses 1 and 2 are different. For Hyp 2, are you going to look at differences in 2 groups (those diagnosed with BDD and those without) on sociocult attitudes? Why? You don’t have a clinical sample, so no reason to assume you would have anyone meet criteria for BDD

Line 116-118 I’m not clear what your second exploratory question is about. Are you looking at the interaction of self-esteem with BDD in predicting SC attitudes?

Line 138  “. . . were approached to participate in . . .”

Line 139-140 “Those students who agreed to participate completed . . .”  (no comma after “students”; delete “at” after “participate”)

Line 148 – Change “civil status” to “marital status” and “current study state” to “current education level”

Use of BDD-YBOCS – not clear why/how this was used. This instrument is supposed to be used for clinical diagnosis, not with a general population. And, it’s administered by clinicians. You had enough clinicians administer the BDD of the YBOCS to 350 participants? This was quite an undertaking! How many clinicians and how did you train them for reliability? Do you have an interrater reliability stat for your clinicians?

Lines 162-163 This may be a problem with your use of commas and semi-colons, but it is unclear how many dimensions you are measuring: BD obsessions (# of items?), BD compulsions (# of items?), avoidance (single item), insight (single item), and total score (12 items). I don’t see BD obsession and compulsion scales used in the literature, so not sure why you divided the scale in this way.

Line 172 Replace “expectances” with “expectations.” However, you really mean that higher scores reflect higher endorsement of societal appearance ideals.

Lines 172-174 Need clarification of the subscales (what they are measuring and # of items). Here is an article that does this nicely: https://jeatdisord.biomedcentral.com/articles/10.1186/2050-2974-1-14

You also need to be sure that the order that you present these subscales is the same order you show them on tables and in discussing them in the text.

Lines 176-178 Follow APA formatting style for punctuation

Line 178-179  “Answers are given on a 4-point rating scale . . . with some items reverse coded,. . .”

Line 182 Comma after “tests,” not after “explored.”

Line 182 – Remove “gender” from the sentence – you did not do a gender by gender chi square. You did two X2 tests to explore if marital status and current education level systematically differed between male and female participants.

Line 197 No comma after “indicate”

Line 198 No comma after “calculate”; a better word for “calculate” is “determine”

Line 199 No comma after “equation”

Line 206 Comma after “those”; no comma after “females”

Line 207 No comma after “single”; You mean “nursing” not “nursery”

Line 211 “. . . correlation coefficients for the body . . . and self-esteem measures.” However, note that Table 1 does not report what your text says it does.

Table 1 should present the M, SD, range, and reliability of your measures. You might also include some text about the intercorrelations of the 4 BDD dimensions (they are moderately to highly correlated, ranging from .23 - .71) and of the 4 SC Attitude scales. This will eliminate these analyses from your tables, making the tables easier to read.

Comment: Your analyses do not follow with your hypotheses in the order you have articulated them. You might consider labeling the analyses sections with the hypothesis. So, Section 3.2 would be Descriptive statistics for study measures, Section 3.3 would be: Hypothesis 1. Associations between BDD and SC attitudes, Section 3.4: Hypothesis 2. Low and high risk of BDD and SC Attitudes, Section 3.5: Hypothesis 3. Relationship of S-E to BDD and SC attitudes (note that you would just be talking about the findings you report on the table for Hypothesis 1), etc.

Hypothesis 1 – BDD and SC attitudes are associated – I anticipate seeing a correlational table of these variables first (Currently your Table 2, but just have the BDD dimensions as the rows and the SC attitudes as the columns. You can add in the S-E correlations (for Hypoth 3) by adding it as both a row and a column to preserve space and then refer to it later when you talk about Hypoth 3).

Hypothesis 2 – Difference between BDD group and non BDD group on SC attitudes comes next (along with your current Table 1)

Hypothesis 3 – Correl of BDD, SC, and S-E (though Hypoth 1 covers this to some extent). I believe the “exploratory” part of this is the addition of SE. So, your next analyses should just discuss correlations found on Table 2 of S-E with BDD and SC.

Hypothesis 4 – to what extent SE influences the assoc btwn BDD and SC attitudes – Couldn’t do this because S-E was not correlated with anything. Need to say this.

Hypothesis 5 – Which dimensions of SC attitudes best predict BDD symptoms.

Lines 218-223 I don’t see any comments on the findings.

Lines 225-226 Table 2 does not report this; Table 2 is a correlation table. It may be that Tables 1 and 2 got switched along the way. Not clear why Age is on this table.

Lines 230-234 Your comments on the findings should just reflect on the hypothesis; the intercorrelations are not relevant here but can go in the Descriptive statistics section. You need to highlight the relevant correlations between the two measures. For example: There were significant, positive relationships for both obsessive thinking about body shape and size as well as compulsive behaviors related to the body concerns and endorsing items that reflect internalization of the thin ideal,  pressure to conform to Western ideals of thinness, and internalization of an athletic ideal body. All relationships reflected medium effect sizes. Additionally, there were small effect sizes in the level of insight participants demonstrated about their body concerns and pressure to conform to ideals of thinness and internalization of an athletic body. Avoidance demonstrated no significant relationships with the Sociocultural Attitude measure.

Line 243. Remove comma after “alter”; end sentence after “covariate.”  New sentence begins: “That is, correlation coefficients . . .”

Lines 246-247 Not a sentence.

Table 3 – What is SCATA? This is the first time you use this acronym. Also, if I read your Table 2 correctly, Athletic ideal body was correlated with the Total BDD score, so not clear why it was excluded in your analyses.

Discussion

My comments will not be as detailed as above, since the whole paper will need to be reworked. There are numerous comma, semi-colon, and run-on sentence problems that I will not correct.

You also overuse the “in our opinion.”

Line 276 You make the claim that Social Networking sites are the reason for concerns about beauty and body shape, but you have no way of knowing this.

The paragraph then goes on to explain why SC attitudes explained so little variance in BDD and what else could be explaining the variance. The first speculation is that social feedback (from where? From whom?) might be the culprit. So, despite that women and men dress modestly, it may be that within social groups (family? Peers) there is pressure to be thin, attractive, etc? I’m not clear what point you are making here.

Similarly, your second point is that there is pressure to find a successful mate and that this is related to attractiveness but not media influences. I don’t see the connection. Where do the ideals of attractiveness come from? Are they different in Iran?

Frankly, for your second hypothesis, you simply don’t have a clinical sample from which to draw any conclusions. You have such a restricted range on the BDD measure as well as the SE measure (these are, after all, successful college students) that you cannot draw any conclusions.

Similarly, trying to determine if SE mediates BDD and SC attitudes is not possible without a range of BDD (Hypoth 3). The literature on SE is large and complex. When it comes to understanding the relationship between SE and body dissatisfaction, psychologists are often flummoxed. Women with high BMI often report liking themselves and feeling satisfied with their lives. There seems to be a bifurcation in our perceptions of self between feeling satisfied with our lives and feeling fat.

Hypothesis 4 needs to be done with Athletic ideal included. Note that SATAQ – General was more predictive of BDD than Media pressure. Why? What is the difference among these scales?

Line 327 – you indicate that this was fully self-rating data. If so, you did something to the BDD that needs to be explained. This is a semi-structured interview that should be done by trained clinicians to assess for pathology.

You have many more limitations to this study (restricted range, high SES, M:F ratio).

Author Response

please see the detailed point-by-point-response

Reviewer 3 Report

The present manuscript, entitled “Sociocultural attitudes towards appearance, self-esteem and symptoms of body-dysmorphic disorders among young adults” aims at evaluating the relationship between BDD symptoms, self-esteem, and sociocultural attitudes towards appearance.

I believe the manuscript is not publishable in its current version; major changes should be performed in order to take into consideration the manuscript for publication.

General feedback:

I have concerns around the novelty of this project; it is unclear how this manuscript extends the literature. Authors should make big efforts to explain the novelty of the study (which is missing in the introduction). Furthermore, the paper has many instances of poor grammar and repetition. Authors must check the manuscript for punctuation and grammar errors. The employed self-report questionnaires (apparently) have not yet been validated in the Iranian context. Finally, the manuscript is characterized by inaccuracy (e. g. mismatch between text and tables).

I think authors should make big efforts to improve the quality of the paper.

Introduction

In general, the introduction does not match with the aims of the study. Authors extensively explain the evolutionary point of view, the waist-to-hip ratio, and the neural areas responsible for the elaboration of beauty and attractiveness. However, no one of these variables were assessed/examined in the study. On the contrary, variables relevant for the study (self-esteem, sociocultural attitudes towards appearance, and BDD) were overlooked in the introduction. Furthermore, authors should explain why they decided to focus on the relationship between BDD symptoms and sociocultural attitudes towards appearance (BDD has never been mentioned in the introduction).

Specific comments:

1) Page 1, line 42: A full stop should be included after the reference. The sentence is too long.

2) Page 2, line 47: Why the study by Kirsch and colleagues (neural processes of human faces and bodies) is relevant in this section of the introduction?.  

3) Page 2, lines 51-69: I do not think that the extensive explanation provided by Authors of  waist-to-hip ratio is relevant for the study. I suggest reducing this section.

3) Page 2, lines 64-69: I do not agree with the conclusion drawn by Authors about beauty and culture. Studies supporting different point of views should be included.

4) Page 2, lines 70-73: Once again, authors focused specifically on the evolutionary point of view and on neural areas to explain why young adults feel under pressure to match society expectation of beauty. They should explain their point in accordance with the variables they assessed in the study.

5) Page 2, line 77: I think authors meant that the use of SNSs was associated with lower body image satisfaction (lower body image has no sense to me).

6) Page 2, lines 80-82: Why do the comment about Pamela Reif’s channel is relevant for the study? I cannot see your point.

7) Page 3, lines 100-103: The link between BDD and Eating Disorders (EDs) should be explain if authors are interested in presenting the similar pattern of associations between BDD, self-esteem, and dysfunctional eating behaviors. I direct the authors to the work conducted by Phillipou et al., 2017, Cerea et al., 2018, and Dingemans et al., 2012. Authors should include these references in their paper.

·         Phillipou, A., Castle, D.J., Rossell, S.L., 2017. Anorexia nervosa: eating disorder or body

·         image disorder? Aust. N. Z. J. Psychiatry 52, 13–14.

·         Cerea, S., Bottesi, G., Grisham, J. R., & Ghisi, M. (2018). Non-weight-related body image concerns and Body Dysmorphic Disorder prevalence in patients with Anorexia Nervosa, 267, 120-125;

·         Dingemans, A.E., van Rood, Y.R., de Groot, I., van Furth, E.F., 2012. Body dysmorphic disorder in patients with an eating disorder: Prevalence and characteristics. Int. J. Eat. Disord. 45, 562–569.

8) Page 3, lines 109-121: The way in which authors explained their hypotheses, study aims, and research questions is confusing. Authors should rephrase this section.

9) Page 3, lines 111-113: Authors assumed that individuals who scored above the cut-off on BDD symptoms reported higher scores of sociocultural attitudes towards appearance. However, no clinical participants with BDD were included in the study. Authors should rephrase this sentence in accordance with their sample.

Methods

1) Procedure: I suggest moving the procedure section after the sample description. Furthermore, authors should specify if participants received a reimbursement for their participation.

2) Sample: In general, no information about participants is provided in this section. Authors should move the first paragraph of Results (3.1 Study participants) within this section. Furthermore, information provided in this section better fit in the procedure section.

Page 3, lines 139-140: Rephrase the sentence. I think it contains a typo.

Page 3, lines 141-143: Authors should explain why being a medicine or a psychology student was one of the inclusion criteria of the research (criterion c).

3) Tools: Authors should include psychometric properties of the employed self-report questionnaires. Furthermore, it is not clear to me if these measures are validated in the Iranian context. If employed measures are not validated in the Iranian context, this information should be included in the limitation section (as a huge limitation).  

Page 4, line 163: How did authors choose the cut-off of 20 for BDD symptoms?

Results

In general, results and tables are imprecise (the description does not match with findings included in tables).

1) Page 5, line 204 (Study sample): This section is descriptive and should be move in the Sample description section.

2) Page 5, line 209: Table 1 does not include correlation coefficients as authors affirmed.

3) Page 6, lines 222-223: The statement: “Higher scores of BDD were associated with higher score of sociocultural attitudes towards appearance” is incorrect. Table 2 displays that the “media as informational source” subscale did not correlate with BDD symptoms. Furthermore, it is not clear to me why authors decided to include age in Table 2. They have never mentioned the role played by age.

4) Page 6, lines 225: Table 2 did not report descriptive statistics of participants at low and high risk of BDD as authors affirmed.

5) Page 7, line 231: Avoidance differed between participants with high and low BDD and the effect sized is Large (shown in table). However, inconsistent information is reported in the text.

6) Page 7, lines 246-247: Authors should include a new paragraph to assess the role played by sociocultural attitude towards appearance dimension in predicting BDD symptoms.

7) Page 7, table 3: Why did authors include age in the regression model? They have never mentioned the role played by age.

Discussion:

In general, authors should explain the novelty and implications of their findings. Furthermore, many provided explanations are not feasible (authors did not assess variables object of the explanation in their study). Limitations should include the university student population and the absence of validated self-report measures.

Pages 8-9, lines 287-291, 297, 308-314: Provided explanations are not in accordance with measures employed in the study (e. g. eating-disordered behaviors were not assessed).

Page 9, line 292: Individuals of the study did not meet BDD criteria (they simply scored above a cut-off).

Page 9, lines 305-308: Authors should rephrase the sentence.

Author Response

(The authors gave the same response as above.)

Round 2

Reviewer 3 Report

Introduction

I think authors improved the quality of the Introduction. However, many grammar mistakes are still present and some sentences are very difficult to follow: Authors must check the manuscript for grammar errors. English revision is needed.

Furthermore, the way authors explained their hypotheses and study aims is still confusing. Authors should make big efforts to make this section more clear.

Specific comments:

1) Page 1, line 41: Please delete the extra space between words.

2) Page 2, lines 73-77: Please provide references for the sentence.

3) Page 2, lines 86-88: Please include commas in the sentence. The sentence is very long and difficult to follow.

4) Page 2, lines 141-157: The way in which authors explained their hypotheses, study aims, and research question is still confusing. Please clearly state your first aim and following hypotheses. Furthermore, clearly state your second aim and following hypotheses (etc.,).

5) Page 3, lines 125-128: The study by Phillipou and colleagues is not explained properly.

Method:

Authors should make bigger efforts to improve the methods section and to provide coherence and clarity.

Procedure:

1) Page 4, line 165: It is not clear to me how experts (trained psychiatrists and clinical psychologists) rated participants’ severity of body dysmorphic disorders. Did experts interview 350 students in person? If this is the case, the procedure section should explain it clearly. 

Results:

Authors should provide coherence in reporting results. Furthermore, results and tables are still imprecise.

1) Page 6, lines 263-264: The sentence is not appropriate for all the emerged results. Please rephrase it.

1) Page 6, lines 259-272: Authors did not report the absence of correlations between avoidance and insight components of the BDD-YBOCS and some of the SATAQ-3 subscales. However, they reported the absence of correlations between SATAQ-3 information subscale and all the BDD-YBOCS components. Please provide coherence when describing your results. Furthermore, pertaining table 1, a value of r = 0.00 is very unusual.

3) Page 6, lines 269-270: The sentence is not appropriate for all the emerged results. Please rephrase it.

4) Page 7, lines 284-285: I don’t see the point of comparing individuals at low risk for BDD with individuals at high risk for BDD in BDD dimension. Indeed, participants have been assigned to groups based on the cut-off of the BDD-YBOCS.

5) Page 7, lines 285-287: Description of the effect sizes does not match with findings included in table 2.

6) Page 7, lines 287: Results about self-esteem are missing.

7) Page 8, lines 296-298: This information is already reported in section 3.2.

8) Page 8, lines 298-299: I suggest including this information is section 3.3 and rephrasing the title of section 3.3.

9) Page 9, line 311: Self-esteem is not included in Table 3.

Discussion:

1) Page 9, lines 325-326: The sentence: “[…] the importance of bodily appearance in relation to body satisfaction […]” is not linked with results of the study. Authors should explain the importance of their results pertaining the relationship between BDD symptoms and sociocultural attitudes toward appearance.

2) Page 9, lines 330-333: Please mention that only some BDD components were associated with sociocultural attitudes toward appearance, and explain why other dimensions were not associated (i.e, avoidance and insight components). Furthermore, a full stop is missing.

3) Page 10, lines 343-345: Please include a reference for the sentence.

4) Page 10, line 355: Explain in which way the second hypothesis was not fully supported.

5) Page 10, line 360: Please correct the typo.

6) Page 11, line 390: Please include the full stop.

7) Page 11, line 419: Please correct the typo

Tables:

1) Table 1: Authors did not include Cronbach’s alphas for avoidance and insight dimensions of the BDD-YBOCS. Please explain why.

Furthermore:

- Table 1 still includes age in the legend; furthermore, the legend is incomplete;

- A bracket for repetitive behaviors is missing.

2) Table 3: This table still includes age in the legend.

Author Response

Please find the detailed point-by-point-response attached as a separate file.

Round 3

Reviewer 3 Report

Please proofread the Manuscript to detect grammar errors.

Author Response

Please find the detailed point-by-point-response attached as a separate file. Thank you for all your kind efforts. 
